# Treatments for gestational diabetes: a systematic review and meta-analysis

Diane Farrar,[1] Mark Simmonds,[2] Maria Bryant,[3] Trevor A Sheldon,[4] Derek Tuffnell,[5] Su Golder,[6] Debbie A Lawlor[7]

[1]Bradford Institute for Health Research, Bradford Royal Infirmary, Bradford, UK
[2]Centre for Reviews and Dissemination, University of York, York, UK
[3]Leeds Institute of Clinical Trials Research, University of Leeds, Leeds, West Yorkshire, UK
[4]Hull York Medical School, University of York, York, UK
[5]Bradford Women's and Newborn Unit, Bradford Teaching Hospitals NHS Foundation, Bradford, UK
[6]Department of Health Sciences, University of York, York, UK
[7]MRC Integrative Epidemiology Unit, School of Social and Community Medicine, University of Bristol, Bristol, UK

**Correspondence to**
Dr Diane Farrar; diane.farrar@bthft.nhs.uk

## ABSTRACT

**Objective** To investigate the effectiveness of different treatments for gestational diabetes mellitus (GDM).

**Design** Systematic review, meta-analysis and network meta-analysis.

**Methods** Data sources were searched up to July 2016 and included MEDLINE and Embase. Randomised trials comparing treatments for GDM (packages of care (dietary and lifestyle interventions with pharmacological treatments as required), insulin, metformin, glibenclamide (glyburide)) were selected by two authors and double checked for accuracy. Outcomes included large for gestational age, shoulder dystocia, neonatal hypoglycaemia, caesarean section and pre-eclampsia. We pooled data using random-effects meta-analyses and used Bayesian network meta-analysis to compare pharmacological treatments (ie, including treatments not directly compared within a trial).

**Results** Forty-two trials were included, the reporting of which was generally poor with unclear or high risk of bias. Packages of care varied in their composition and reduced the risk of most adverse perinatal outcomes compared with routine care (eg, large for gestational age: relative risk 0.58 (95% CI 0.49 to 0.68; $I^2$=0%; trials 8; participants 3462). Network meta-analyses suggest that metformin had the highest probability of being the most effective treatment in reducing the risk of most outcomes compared with insulin or glibenclamide.

**Conclusions** Evidence shows that packages of care are effective in reducing the risk of most adverse perinatal outcomes. However, trials often include few women, are poorly reported with unclear or high risk of bias and report few outcomes. The contribution of each treatment within the packages of care remains unclear. Large well-designed and well-conducted trials are urgently needed.

**Trial registration number** PROSPERO CRD42013004608.

### Strengths and limitations of this study

► This systematic review evaluates available interventions for the treatment of gestational hyperglycaemia and includes a network meta-analysis comparing all pharmacological treatments for gestational diabetes.

► A large number of trials conducted in varied populations have been included.

► For some comparisons, the numbers of trials included were few, and outcomes reported were few.

► Trial quality was generally poor with subsequent high or unclear risk of bias.

## INTRODUCTION

Treatment of gestational diabetes mellitus (GDM) aims to reduce hyperglycaemia and in turn reduce the risk of adverse perinatal outcomes including large for gestational age (LGA), macrosomia, shoulder dystocia, neonatal hypoglycaemia and the need for caesarean section. Diet modification is often used as first-line treatment, and if partly or wholly unsuccessful or where women have substantially elevated glucose at diagnosis, pharmacological treatments (metformin, glibenclamide (glyburide) and/or insulin) are offered.

Previous systematic reviews have investigated the effectiveness of treatments for GDM.[1–15] Although results from these reviews generally indicate that treatment reduces the risk of adverse perinatal outcomes, the searches have variable inclusion criteria and were undertaken between 2009[1 5] and 2014[2–4 6–8 10 11 16 16] with three reviews with searches in 2015,[9 14 15] and since then, several trials have been published and recommended criteria for GDM diagnosis have changed. Some reviews have included observational studies, and most do not review all treatments, with the exception of the Cochrane treatments review[1] (which is now out of date and has been divided for future updates) and the UK National Institute for Health and Care Excellence (NICE) guideline.[16] Consequently, most previous reviews do not provide an assessment of all available treatments, and most have not used a network meta-analysis to determine the most effective pharmacological treatment across all alternatives included in any randomised controlled trial (RCT).

The aim of this study was to systematically review and, where appropriate, pool all results from RCTs of the effect of any treatment on GDM and to determine which treatment is the most effective.

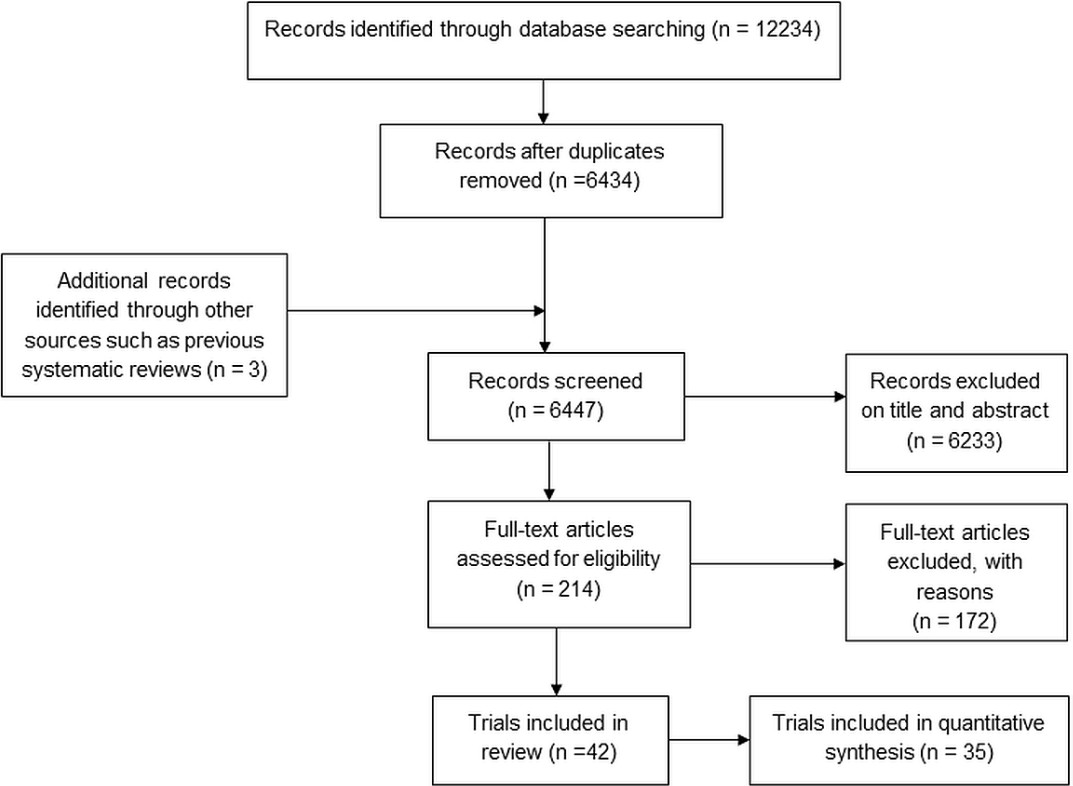

**Figure 1** Search process.

## METHODS

We conducted a systematic review, meta-analysis and network meta-analysis to evaluate whether treatments for GDM reduce the risks of adverse perinatal outcomes and to compare the effectiveness of these treatments.

This review and meta-analysis was conducted in accordance with Cochrane systematic reviews[17] and the Centre for Reviews and Dissemination recommendations[18]; we have reported our findings following the Preferred Reporting Items for Systematic Reviews and Meta-Analyses (PRISMA) reporting guidelines (see online supplementary research checklist).[19] This review forms part of a larger health technology assessment report of the diagnosis and management of GDM.[20]

### Patient involvement

The outcomes we included were from the Cochrane Pregnancy and Childbirth Group's standardised outcomes for reviews of diabetes in pregnancy. Women who had experienced or had the potential to experience GDM contribute to the design and appraisal of this group's methods and reviews and therefore have influenced the design of this review and outcomes examined.[21]

### Search methods

The search strategies were designed to identify records of RCTs of treatment of women with GDM, added to search sources since the search date (July 2011, trials awaiting classification) of the Cochrane 'treatments for

GDM' review.[1] The bibliographic databases searched were MEDLINE and MEDLINE in Process, Embase and the Cochrane Central Register of Controlled Trials. Strategies were not restricted by language and were developed using a combination of subject indexing terms and free text search terms in the title and abstract fields. Searches were first conducted in September 2013 and updated in October 2014 and 6 July 2016, using the same search strategies. Information on studies in progress was sought by searching relevant trial registers including Clinical-Trials.gov.

We also searched previously published systematic reviews to ensure any eligible RCTs from these were included in our review if eligible.[2–9] In addition, we checked the references of included journal articles. An example of search terms for MEDLINE is included in online supplementary file 1.

### Study selection
#### Inclusion and exclusion criteria

We included RCTs in which women with diagnosed GDM or impaired glucose tolerance (IGT) (using any definition) were randomised to a treatment designed to lower blood glucose (pharmacological or dietary modification) compared with routine antenatal care (however defined by the trial) or another treatment. Trials including women with pre-existing diabetes were excluded. Trials had to report effects on adverse perinatal outcomes. Included

outcomes (defined in any way by the trials) were gestational age at birth, birth weight (BW), macrosomia, LGA, shoulder dystocia, preterm birth (less than 37 weeks gestation), neonatal hypoglycaemia, admission to neonatal intensive care unit (NICU), caesarean section (elective or emergency), pre-eclampsia, pregnancy-induced hypertension (PIH), induction of labour, instrumental birth (forceps or ventouse), Apgar score at 5 min and negative treatment effects (eg, gastrointestinal upset, well-being). Data on side effects and quality of life measures were also examined. Conference abstracts and letters to journals were eligible for inclusion if they reported sufficient outcome data.

## Data extraction and risk of bias assessment

Title and abstract screening and then full-text screening were performed by two reviewers (DF, MS, MB or SG) with disagreements resolved by consensus or by the third reviewer. The risk of bias of the included trials was assessed using the Cochrane risk of bias tool,[22] which considers sequence generation, allocation concealment, blinding of participants and medical staff to treatment allocation, blinding of assessors, loss to follow-up, selective reporting of outcomes and other sources of bias. Each criterion was classified as being at low or high risk of bias or unclear. Two reviewers independently assessed all criteria (DF, MS or SG).

## Statistical analysis

Trials were divided into categories according to the following included treatments: (1) insulin versus metformin; (2) insulin versus glibenclamide (glyburide); (3) metformin versus glibenclamide; (4) packages of care: diet or dietary advice with or without exercise or glucose monitoring, with or without supplemental metformin, glibenclamide or insulin, compared with routine antenatal care; and (5) comparisons of different dietary modifications.

For dichotomous outcomes, the relative risk (RR) comparing each group, with its 95% CI, was calculated from the numbers of outcome events in each randomised group and the number randomised to each group. For continuous outcomes, the difference in means between groups was calculated from the mean and SD of the outcome. For each outcome and within each of the treatment categories, RRs or differences in means were pooled in random-effects DerSimonian-Laird meta-analyses.[23] Heterogeneity was assessed using $I^2$.[24] Analyses were performed to investigate differences in risk of outcomes across varying degrees of hyperglycaemia (defined by a positive/negative GDM screening and diagnostic test). Because of the large number of treatments and outcome comparisons, pooled estimates only are presented in the main paper. Tests for publication bias were considered, but not performed, because there were insufficient trials in any meta-analysis for such tests to be reliable.

We also conducted a network meta-analysis to combine information across multiple treatments simultaneously;

this combines direct and indirect data to improve the estimation of the effectiveness of treatments and specifically to try to estimate which is the most effective of a number of different treatment options.[25–28] Analyses were undertaken for each dichotomous outcome using a Bayesian approach, based on the models originally created by Lu and Ades,[29] using the OpenBUGS[30] software. The model has a 'binominal-normal' structure; that is, events were assumed to follow a binomial distribution, with log odds and random effects being normally distributed. Vague normal priors (mean 0, variance 10 000) were used except for heterogeneity, where an inverse-gamma (0.1, 0.1) distribution was used. The model fit and consistency were assessed by comparing the results to the meta-analyses comparing each treatment directly.

Each model generated a comparison between treatments, expressed as an OR and as a percentage indicating the probability that the treatment was the best treatment to reduce the incidence of the adverse outcome. ORs were used to ensure model stability because log ORs more closely follow a normal distribution than RRs. The probabilities of being most effective treatment were calculated from the posterior odds as part of the Bayesian model developed by Lu and Ades.[29] This approach was not possible for continuously measured outcomes and so was not undertaken for gestational age, BW and Apgar score. As there were no trials comparing diet modification to pharmacological treatments, diet modification could not be included in the network meta-analyses.

## RESULTS

### Details of included and excluded trials

A total of 12 234 citations were identified by the original and the two update searches. These citations were combined with three additional citations identified by previous systematic reviews conducted prior to our first searches.[1–5] Following de-duplication and inclusion of additional records, 6437 citations were reviewed. Of these, 214 were judged potentially eligible based on title and abstract. After obtaining the full-text publications and assessing eligibility, 42 trials were included, and 35 of these were combined in at least one meta-analysis (figure 1).

Having extracted data from the RCTs assessing packages of care and dietary intervention comparisons (table 1), we decided that it was not appropriate to pool results from trials comparing dissimilar dietary modification interventions (table 1). Packages of care included various combinations of interventions; however, all packages of care compared with routine care trial results were pooled in meta-analyses.

We included eight publications not included in any previous published review. One compared metformin and insulin[31]; one, glibenclamide and insulin[32]; four, packages of care with routine care[33–36]; and two compared different dietary modification interventions.[37 38] Six of these trials were reported after the

**Table 1** Trials comparing a package of care starting with dietary modification to routine care and trials comparing a dietary modification with another dietary modification

Trials comparing a package of care (starting with dietary modification) to routine care

| First author | Year | Location | No | Screening strategy used to determine need for diagnostic test | Diagnostic test and glucose thresholds used to diagnose GDM (mmol/L) | Intervention group | Control group | Insulin use in diet group | In meta-analyses | Meta-analysis outcome |
|---|---|---|---|---|---|---|---|---|---|---|
| Bevier[45] | 1999 | USA | 103 | 50 g OGCT >7.8 | Positive OGCT, negative 100 g OGTT, levels not reported | Dietary counselling and home monitoring | Routine care | If needed | Yes | Apgar 5 min, BW, C-section, GA at birth, induction, instrumental birth, macrosomia, pre-eclampsia, shoulder dystocia |
| Bonomo[46] | 2005 | Italy | 300 | Risk factors and 50 g OGCT | Positive OGCT >7.8, negative 100 g OGTT 'C&C criteria' | Dietary advice and monitoring | Routine care | Not reported | Yes | Apgar 5 min, BW, C-section, GA at birth, LGA, macrosomia, NN hypoglycaemia, NICU admission |
| Crowther[47] | 2005 | UK/Australia | 1000 | Risk factors or 50 g OGCT | 75 g OGTT fasting <7.8 and 2 hours >7.8 and <11.1 | Individualised dietary advice, monitoring and pharmacological treatments | Routine care | If needed | Yes | Apgar 5 min <7, BW, C-section GA at birth, induction, macrosomia, NN hypoglycaemia, NICU admission, pre-eclampsia, shoulder dystocia |
| Deveer[33] | 2013 | Turkey | 100 | Universal 50 g OGCT >7.8 and <10.0 | Positive OGCT, negative 100 g OGTT fasting <5.3, 1 hour <10.0, 2 hours <8.8 and 3 hours <7.8 | Calorie diet | Routine care | Not reported | Yes | BW, C-section, GA at birth, LGA, macrosomia, NICU admission, pre-eclampsia, preterm birth |
| Elnour[48] | 2006 | UAE | 180 | Not reported | 100 g OGTT, 'C&C criteria' | Diet education, exercise, monitoring and pharmacological treatments | Routine care | If needed | Yes | C-section, LGA, macrosomia, NN hypoglycaemia, NICU admission, pre-eclampsia, preterm birth, shoulder dystocia |
| Fadl[34] | 2015 | Sweden | 66 | Risk factors | 75 g OGTT<7.0, >10.0, <12.2 | Diet education, exercise, monitoring and pharmacological treatments | Routine care | If needed in intervention group only | Yes | BW, C-section, LGA, GA at birth, macrosomia, pre-eclampsia, instrumental birth, induction, NICU admission |

Continued

**Table 1** Continued

| First author | Year | Location | No | Screening strategy used to determine need for diagnostic test | Diagnostic test and glucose thresholds used to diagnose GDM (mmol/L) | Intervention group | Control group | Insulin use in diet group | In meta-analyses | Meta-analysis outcome |
|---|---|---|---|---|---|---|---|---|---|---|
| Garner[49] | 1997 | Canada | 299 | 75 g OGCT >8.0 | 75 g OGTT fasting >7.5 and 2 hours >9.6 | Dietary counselling, restricted calorie intake, monitoring and insulin if required | Routine care | If needed | Yes | BW, C-section, GA at birth, macrosomia, NN hypoglycaemia, pre-eclampsia, preterm birth, shoulder dystocia |
| Landon[50] | 2009 | USA | 958 | 50 g OGCT >7.5 to <11.1 | 100 g OGTT fasting <5.3, 2 or more, 1 hour >8.6 or 2 hours >8.6 | Individualised dietary advice, monitoring and insulin | Routine care | If needed | Yes | BW, C-section, GA at birth, induction, macrosomia, NN hypoglycaemia, NICU admission, pre-eclampsia, preterm birth, shoulder dystocia |
| Li[51] | 1987 | Hong Kong | 58 | Risk factors | 100 g OGTT, two or more: fasting >5.8, 1 hour >10.6, 2 hours >9.2, 3 hours >8.1, then 75 g OGTT fasting <8.0 or 2 hours <11.0 | 30–35 g/kg carbohydrate diet and monitoring | Routine care | Not reported | Yes | BW, C-section, GA at birth, induction, macrosomia |
| O'Sullivan[52] | 1966 | USA | 615 | OGCT or risk factors | 100 g OGTT two or more fasting >6.1, or 1 hour >9.1 or 2 hours >6.7 or 3 hours >6.1 | Low-calorie diabetic diet | Standard diabetic diet | Only in intervention group | Yes | Macrosomia, preterm birth |
| Yang[35] | 2003 | China | 150 | Not reported | Not reported | 'Intensive' diabetes management | Routine care | If needed | Yes | C-section, shoulder dystocia |
| Yang[36] | 2014 | China | 700 | | 75 g OGTT fasting 5.1, 1 hour 10.0, 2 hours 8.5 | Individual and group dietary/physical intervention | Routine care | If needed | Yes | BW, C-section, GA at birth, induction, macrosomia, NN hypoglycaemia, PIH, pre-eclampsia, preterm birth, shoulder dystocia |
| Trials comparing a dietary modification with another dietary modification | | | | | | | | | | |
| Asemi[53] | 2014 | Iran | 52 | 50 g OGCT | OGCT >7.8, 75 g OGTT fasting >5.1, 1 hour >10.0, 2 hours >8.5 | DASH diet | Control diet | Women with GDM excluded, therefore insulin not required | No | – |
| Cypryk[54] | 2007 | Poland | 30 | Not reported | Levels not reported only that the WHO criteria were used | High-carbohydrate diet | Low-carbohydrate diet | If needed | No | – |

Continued

**Table 1** Continued

| First author | Year | Location | No | Screening strategy used to determine need for diagnostic test | Diagnostic test and glucose thresholds used to diagnose GDM (mmol/L) | Intervention group | Control group | Insulin use in diet group | In meta-analyses | Meta-analysis outcome |
|---|---|---|---|---|---|---|---|---|---|---|
| Louie[55] | 2011 | Australia | 99 | Not reported | 75 g OGTT ≥5.5, 1 hour >10.0 or 2 hours >8.0 | Low-GI diet | High-fibre moderate-GI diet | If needed | No | – |
| Ma[37] | 2015 | China | 83 | 50 g OGCT | 75 g OGTT ≥5.8, 1 hour >10.6, 2 hours >9.2 or 3 hours 8.1 | Low glycaemia load diet | Usual diet | If needed* | No | – |
| Moreno-Castilla[56] | 2013 | Spain | 152 | 50 g OGCT >7.8 | 100 g OGTT >5.8, 1 hour >10.6, 2 hours >9.2, 3 hours >8.1 | Low-carbohydrate diet | Control diet | If needed | No | – |
| Rae[57] | 2000 | Australia | 124 | Not reported | (Glucose load not reported) OGTT fasting >5.4 or 2 hours >7.9 | Calorie-restricted diet | Usual diet | If needed | No | – |
| Yao[38] | 2015 | China | 33 | 50 g OGCT fasting >5.8 'post-load' >7.8 | 100 g OGTT fasting >5.3, 1 hour >10.0, 2 hours >8.6, 3 hours >7.8 | DASH diet | Usual diet | If needed | No | – |

*Women who required insulin were excluded from the trial's analyses.
BW, birth weight; C-section, caesarean section; DASH diet, dietary approaches to stop hypertension; GA; gestational age; GDM, gestational diabetes mellitus; LGA, large for gestational age; NICU, neonatal intensive care unit; NN, neonatal; OGCT, oral glucose challenge test; OGTT, oral glucose tolerance test; PIH, pregnancy-induced hypertension.

search dates of the previous reviews and were published in 2014 or 2015; the remaining two trials (dietary modification interventions or packages of care) did not fulfil other review's inclusion criteria. Few trials reported side effects or measures of participant satisfaction or well-being.

Trials generally included women with GDM diagnosed following a 75 or 100 g oral glucose tolerance test (OGTT) using a variety of international[39–41] and locally[42 43] recommended thresholds; although some included women with 'mild or borderline' GDM (positive oral glucose challenge test (OGCT), negative OGTT) and others included women with IGT, current diagnostic criteria[16 44], however, may now consider these women as having GDM rather than a separate and milder condition.

### Quality – risk of bias assessment

Overall, reporting of and many aspects of trial quality were poor with the result that risk of bias was generally unclear or high (online supplementary table 1). The randomisation procedure and group allocation were rarely described, although all trials reported that participants were 'randomised'. Blinding of participants, medical staff and outcome assessors was generally not reported, but as most trials include some additional intervention above routine care such as diet advice or a pharmacological treatment, it is probable that participants and most clinicians could not be blinded, although outcome assessment could have been. Most trials had reasonably complete outcome data and loss to follow-up was low, although for some trials, analysis was not conducted on an intention-to-treat basis (so the analysis did not include all women randomised). Selective reporting was assessed as minimal, as the majority of trials presented results for all prespecified outcomes (the possibility that some trials collected data on outcomes but did not report them cannot be ruled out however).

Generally, women were eligible for inclusion in trials evaluating pharmacological treatments if they were unable to achieve adequate glycaemic control with dietary and lifestyle management. Therefore, there is the possibility that those included may have had more severe or refractory hyperglycaemia or may adhere less well to lifestyle interventions than those women who did not require pharmacological treatments to control hyperglycaemia. The specific criteria for the addition of supplemental insulin in trials were often not reported, although some trials did report that supplemental insulin was prescribed if 'glycaemic control was not achieved by participants'. It is probable that thresholds for what is defined as 'good' control differed between trial centres (if multisite) and trials.

### Packages of care and dietary modification trials

Twelve trials evaluated a package of care (a combination of treatments starting with dietary modification and/or exercise and/or monitoring and/or supplemental pharmacological treatments) (table 1)[33–36 45–52] compared with

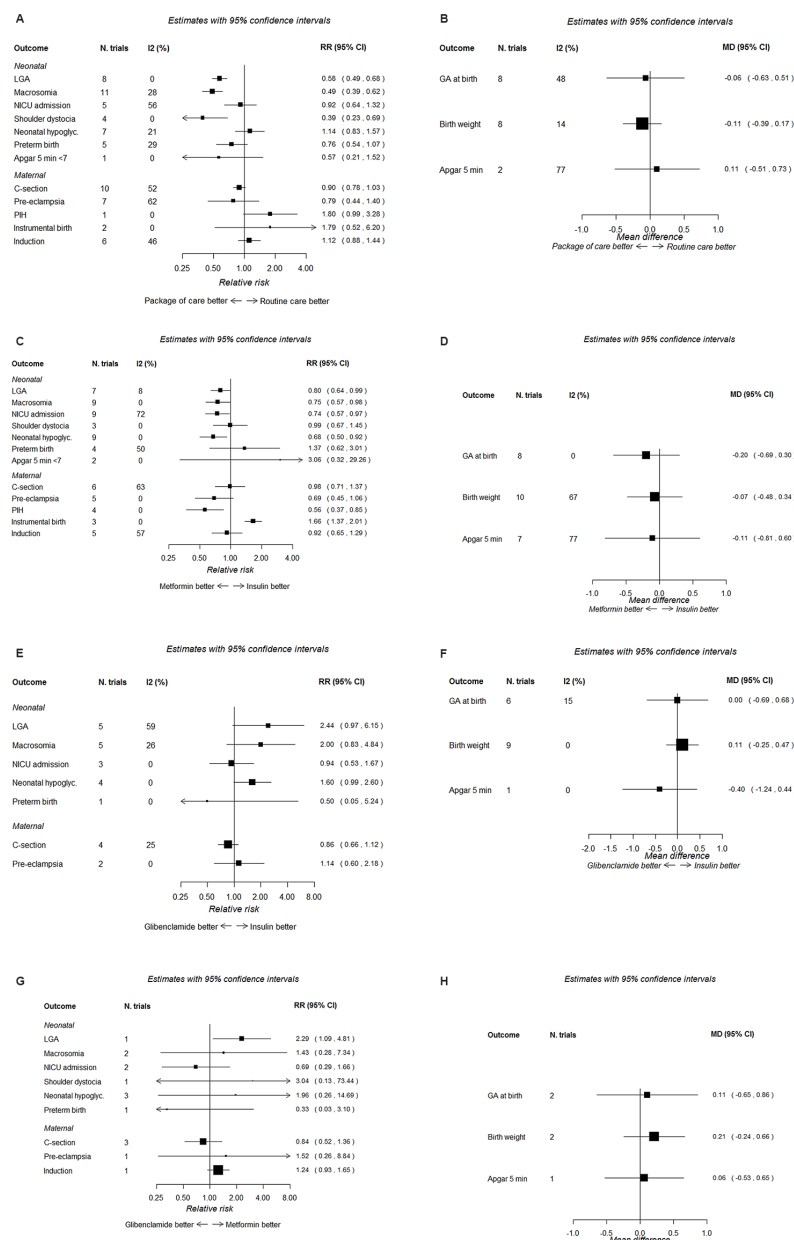

**Figure 2** Forest plots for treatment comparisons and perinatal outcomes. (A) Packages of care (starting with dietary modification) versus routine care: dichotomous outcomes. (B) Packages of care (starting with dietary modification) versus routine care: continuous outcomes. (C) Metformin versus insulin: dichotomous outcomes. (D) Metformin versus insulin: continuous outcomes. (E) Glibenclamide versus insulin: dichotomous outcomes. (F) Glibenclamide versus insulin: continuous outcomes. (G) Glibenclamide versus metformin: dichotomous outcomes. (H) Glibenclamide versus metformin: continuous outcomes.

routine care. Data from these 12 trials are combined in at least one meta-analysis (figure 2A,B).

Seven trials[37 38 53–57] evaluated a variety of dietary modifications and compared them to other dietary modifications (table 1). The composition of each dietary modification was generally well reported; however, the interventions and comparisons were too diverse to allow pooling of data. There was no evidence that one type of dietary modification was superior over another, although trials included few women (online supplementary figures 1 and 2). None of these seven trials reported side effects or quality of life measures.

The composition of the dietary modification was poorly reported in the 'packages of care' trials (the 12 trials included in the meta-analyses). Overall (in all packages of care and dietary modification trials), 10 out of 19 trials reported that insulin was provided if required; in one trial, insulin was only provided if needed in the intervention group; and for the remainder, it was unclear or not reported if supplemental insulin was provided. The screening and diagnostic tests, criteria and glucose

thresholds used to define GDM (and included/exclude women in the trials) varied across the trials (table 1). For the meta-analysis, the varying forms of dietary modification and/or pharmacological treatment use were not examined.

Packages of care (starting with dietary modification and possibly including monitoring and pharmacological interventions) reduced the risk of shoulder dystocia by 60%, LGA and macrosomia by around 50%, pre-eclampsia by 20% and the incidence of caesarean section by 10% compared with routine care (figure 2A), although for pre-eclampsia and caesarean section, the CIs included the null value. BW was reduced by approximately 110 g in the packages of care compared with routine care group (figure 2B). The degree of heterogeneity ($I^2$) varied by outcome from 0% to 77%. No 'packages of care trial' reported side effects; two trials reported quality of life scores[47 48] indicating higher (better) quality of life scores for women in the intervention compared with the routine care group.

### Trials comparing metformin with insulin
Eleven trials compared metformin with insulin (table 2).[31 43 58–66] However, most trials reported supplemental insulin use in the metformin group with the exception of two trials.[31 64] The risk of most outcomes, including LGA, macrosomia, NICU admission, neonatal hypoglycaemia, pre-eclampsia, PIH and induction of labour, was lower in those randomised to metformin rather than insulin; instrumental delivery was greater in those randomised to insulin (figure 2C). BW, gestational age and Apgar score as continuous measurements did not differ notably between the two treatments (figure 2D). Six trials reported the proportion of women with metformin-associated gastrointestinal upset (between 4% and 46%).[58–60 63 65 66] No trial reported quality of life measures.

### Trials comparing glibenclamide (glyburide) with insulin
Nine trials compared glibenclamide with insulin (table 3).[32 67–74] Figure 2E shows the RRs of dichotomous outcomes, suggesting that insulin may be relatively more effective than glibenclamide in reducing the risk of several adverse outcomes; CIs are wide and include the null value however. There was no difference between insulin and glibenclamide for continuous outcomes (figure 2F). One trial reported that glibenclamide was associated with side effects in 3/48 (6%) of women.[72] No trial reported quality of life measures.

### Trials comparing glibenclamide (glyburide) with metformin
Only three trials were identified that directly compared glibenclamide with metformin, and these were relatively small trials including between 149 and 200 women (table 4).[75–77] Figure 2G shows the risk of dichotomous, and figure 2H shows continuous outcomes. These suggest that metformin is more effective at reducing risk of LGA and possibly macrosomia. However, for several of the

outcomes (eg, LGA), only data from one of these trials are available; it is therefore not possible to make robust conclusions about the relative benefits of metformin and glibenclamide from these direct comparisons. No trials reported side effects or quality of life measures.

### Network meta-analysis comparing glibenclamide (glyburide), insulin and metformin
Figure 3 shows the relationship of treatment comparisons, and table 5 shows the estimated probability of a treatment being the most effective at reducing the risk of each dichotomous outcome. Only dichotomous outcomes reported in at least two glibenclamide trials (either in comparison to insulin or metformin) were included in these analyses to ensure that there were sufficient trials (and participants) included. When all three treatments are jointly compared, these analyses suggest that, for all outcomes, with the exception of caesarean section, metformin is most likely to be the most effective treatment, with its probability of being most effective in reducing risk being 96.3%, 94.0%, 92.8%, 84.0% and 61.2%, respectively, for neonatal hypoglycaemia, macrosomia, LGA, pre-eclampsia and admission to NICU (the probability of being most effective for reducing risk of caesarean section was 9.7% for metformin, glibenclamide was most likely to be most effective at reducing the risk of caesarean section (79.9%)). The results of the network meta-analysis (figure 4) are consistent with the direct comparisons between treatments shown in figure 2A–H, suggesting that metformin is more effective than insulin or glibenclamide at reducing the majority of adverse outcomes. However, many of these comparisons are based on small numbers and have wide CIs that sometimes include the null value.

### DISCUSSION
The key finding of our review is that, despite understanding of hyperglycaemia/GDM and its relationship to adverse perinatal outcomes having existed for at least seven decades[78] and 42 RCTs completed on its treatment, trials are still being conducted that are of limited size and of poor quality (with subsequent unclear or high risk of bias), and therefore, which treatment is the most effective remains unclear. Given the changing characteristics of the population and the lower fasting diagnostic threshold (compared with previous criteria)[40] recommended by the International Association of Diabetes and Pregnancy Study Groups (IADPSG)[44] and UK NICE,[16] it is important to understand how treatments affect outcomes for these women. Trials do not always report GDM diagnostic criteria clearly, and this is important considering the potential influence on GDM population size and the magnitude of effect.[16 44] Our detailed review, including only evidence from RCTs, provides some support for a 'step up approach' in the treatment of hyperglycaemia, from dietary interventions, through addition of metformin (in preference to glibenclamide

on

**Table 2** Trials comparing metformin to insulin

| First author | Year | Location | No | Diagnostic test and glucose thresholds used to diagnose GDM | Screening strategy* | Meta-analysis outcome |
|---|---|---|---|---|---|---|
| Ainuddin[66] | 2014 | Pakistan | 150 | 75 g OGTT two or more; fasting 5.3, 1 hour 10.0, 2 hours 8.6 | 50 g OGCT ≥7.8 | PIH, pre-eclampsia, GA at delivery, induction, C-section, LGA, NICU admission, neonatal hypoglycaemia |
| Hague[64] | 2003 | Australia | 30 | 75 g OGTT fasting >5.5 or 2 hours >8.0 | Risk factors | BW, pre-eclampsia, GA at birth, induction, C-section, macrosomia, hypoglycaemia |
| Hassan[65] | 2012 | Pakistan | 150 | 75 g OGTT two or more levels fasting >5.3, 1 hour >10.0 or 2 hours >8.6 | 50 g OGCT >7.8 | Apgar 5 min, GA at birth, induction, C-section, BW, macrosomia, hypoglycaemia, NICU admission |
| Ijas[63] | 2010 | Finland | 100 | 75 g OGTT fasting >5.3, 1 hour >11.0 or 2 hours >9.6 | Risk based | Apgar 5 min, BW, C-section, GA at birth, induction, instrumental birth, LGA, macrosomia, hypoglycaemia, NICU admission |
| Mesdaghinia[62] | 2013 | Iran | 200 | 100 g OGTT two or more; fasting >5.3 or 1 hour >10.0 or 2 hours >8.6 or 3 hours >7.8 | 50 g OGCT – levels not reported | BW, macrosomia, LGA, hypoglycaemia, NICU admission, shoulder dystocia, 5 min Apgar <7, preterm birth |
| Moore[61] | 2007 | USA | 63 | 100 g OGTT two or more; fasting >5.8 or 1 hour >10.5 or 2 hours >9.1 or 3 hours >8.0 | 50 g OGCT >7.8 | Apgar 5 min, BW, macrosomia, hypoglycaemia, NICU admission |
| Niromanesh[60] | 2012 | Iran | 160 | 100 g OGTT two or more fasting >5.3, 1 hour >10.0, 2 hours, 3 hours >8.6 or 3 hours >7.8 | 50 g OGCT >7.2 | Apgar 5 min, pre-eclampsia, PIH GA at birth, induction, C-section, shoulder dystocia, BW macrosomia, LGA, NICU admission, hypoglycaemia, preterm birth |
| Rowan[59] | 2008 | Australia / NZ | 751 | 75g OGTT fasting >5.5 or 2 hours >8.0 | Risk factors | Apgar 5 min <7, BW, GA at birth, LGA, NICU admission, PIH, pre-eclampsia, preterm birth |
| Spaulonci[58] | 2013 | Brazil | 94 | 75 g or 100 g OGTT fasting >5.3 or 1 hour >10.0 or 2 hours >8.0 and two or more fasting >5.3, 1 hour >10.0, 2 hours, 3 hours >8.6 or 3 hours >7.8, respectively | No screening | GA at birth, BW, Apgar 5 min, macrosomia, hypoglycaemia, pre-eclampsia, preterm birth, C-section |
| Tertti[43] | 2013 | Finland | 217 | 75 g OGTT both criteria: fasting ≥4.8, 1 hour ≥10.0, 2 hours ≥8.7 and fasting ≥5.3, ≥10.0 and ≥8.6, respectively | Risk factors | GA at birth, BW, Apgar at 5 min, induction, instrumental birth, C-section, LGA, macrosomia, preterm birth, PIH, pre-eclampsia, NICU admission, hypoglycaemia |
| Zinnat[31] | 2013 | Bangladesh | 450 | Not reported† | Not reported† | Macrosomia, shoulder dystocia, C-section, instrumental birth hypoglycaemia, NICU admission |

*It is assumed unless otherwise reported that the screening strategy advocated by the criteria used was adhered to.

†Conference abstract.

BW, birth weight; C-section, caesarean section; GA, gestational age; GDM, gestational diabetes mellitus; LGA, large for gestational age; NICU, neonatal intensive care unit.

**Table 3** Trials comparing glibenclamide (glyburide) to insulin

| First author | Year | Location | No | Diagnostic test and glucose thresholds used to diagnose GDM | Screening strategy* | Outcome |
|---|---|---|---|---|---|---|
| Anjalakshi[67] | 2007 | India | 23 | 75 g OGTT 2 hours >7.8 | Universal OGTT | BW |
| Bertini[68] | 2005 | Brazil | 70 | 75 g OGTT fasting >6.1 or 2 hours >7.8 | Not reported | BW, C-section, Apgar 5 min, GA at birth, LGA |
| Lain[69] | 2009 | USA | 99 | 100 g OGTT two or more: fasting >5.3, 1 hour >8.6 or 2 hours >8.6 | 50 g >7.5 | BW, GA at birth, LGA, macrosomia |
| Langer[70] | 2000 | USA | 404 | 100 g OGTT fasting >5.3 to <7.8 | 50 g OGCT>7.3 | BW, C-section, GA at birth, LGA, macrosomia, hypoglycaemia, NICU admission, pre-eclampsia |
| Mirzamoradi[32] | 2015 | Iran | 96 | Glucose load not reported; OGTT two or more: fasting >5.3, 1 hour >10.0, 2 hours >8.3 | Universal OGTT | BW, C-section, GA at birth, NICU admission, hypoglycaemia, pre-eclampsia |
| Mukhopadhyay[71] | 2012 | India | 60 | 75 g OGTT 2 hours >7.8 | No screening | BW, GA at birth, LGA, hypoglycaemia |
| Ogunyemi[72] | 2007 | USA | 97 | Not reported | Not reported | BW, C-section, GA at birth, hypoglycaemia, |
| Silva[73] | 2007 | Brazil | 68 | 75 g OGTT fasting >6.1 or 2 hours >7.8 | No screening | BW, C-section, LGA, macrosomia, |
| Tempe[74] | 2013 | India | 64 | 100 g OGTT two or more: fasting >5.3, 1 hour >10.0, 2 hours >8.6 or 3 hours >7.8 | 50 g OGCT >7.2 | BW, GA birth, macrosomia, hypoglycaemia, NICU admission, pre-eclampsia, preterm birth |

*It is assumed unless otherwise reported that the screening strategy advocated by the criteria used was adhered to.
BW, birth weight; C-section, caesarean section; GA, gestational age; GDM, gestational diabetes mellitus; LGA, large for gestational age; NICU, neonatal intensive care unit.

(glyburide)) through addition of insulin. Considering that hyperglycaemia in pregnancy has various causes and many women will be treated successfully with diet and lifestyle interventions (because lower thresholds lead to less severe hyperglycaemia being classified as GDM), using an integrated individual approach to its management is likely to work best, although trials and reviews continue to be conducted that pay little attention to the influence of non-pharmacological treatments for GDM and often do not provide information on the severity of hyperglycaemia in treatment groups.

We have taken a pragmatic approach to evaluating the many trials examining treatment packages of care for women diagnosed with hyperglycaemia/GDM so that our results will be generalisable to most clinical situations. Several previous reviews have focused exclusively on pharmacological treatments[2 6 8 9 12–15]; however, others have also suggested that packages of care with a 'step up' approach are the most effective.[1 3–5] The severity of hyperglycaemia may influence the effectiveness of a treatment; however, many trials do not report treatment subgroup baseline glycaemic levels (eg, diet only, diet and metformin or insulin, or metformin with supplementary insulin).[34–36 45 47 48 51 62–65 79] For those trials reporting baseline glycaemic levels by treatment subgroup, there is inconsistency, with some reporting significant differences between groups[59 66] and others reporting no difference.[43 58 60] Understanding of treatment effects would be improved if baseline OGTT levels were presented by treatment subgroup in future trials.

The number of trials and women included in previous reviews varies. One recent review had broadly similar inclusion criteria to ours, comparing any package of care for the treatment of GDM with no treatment (routine care) and included five trials with 2643 women.[3] Our review includes all these trials, plus a further seven (included in the meta-analysis) increasing the number of women to 4512 and indicating that RCTs in this area continue to be conducted, but not with the size or quality that allows us to have a robust evidence base for the treatment of GDM in a contemporary population. Pooled estimates are generally consistent across reviews of packages of care irrespective of the number of trials included because estimates are driven in all reviews by the two largest, which are also the highest quality trials; however, these trials were conducted in populations using diagnostic criteria that would provide populations with more severe hyperglycaemia (and therefore the potential for a larger effect size).[47 50] For example, our analysis shows the risk of macrosomia is halved when a package of care is provided compared with routine care (11 trials, RR 0.49, 95% CI 0.39 to 0.62), confirming estimates from the most recent previous review (RR 0.50, 95% CI 0.35 to 0.71).[3] These two large and well-conducted RCTs were published in 2005 and 2009,[47 50] and since then, several smaller and poorer quality trials have been published. These two previous large well-conducted trials cannot provide precise estimates of effect on the wider range of

**Table 4** Trials comparing glibenclamide to metformin

| First author | Year | Location | No | Diagnostic test and thresholds used to diagnose GDM (mmol/L) | Screening strategy[a] | Outcome |
|---|---|---|---|---|---|---|
| George[76] | 2015 | India | 159 | 100 g OGTT two or more; fasting >5.3 or 1 hour >10.0 or 2 hours >8.6 | Not reported | BW, GA at birth, macrosomia, hypoglycaemia |
| Moore[75] | 2010 | USA | 149 | 100 g OGTT two or more; fasting >5.3 or 2 hours >6.7 | 50 g OGCT >7.2 | BW, C-section, GA at birth, macrosomia, hypoglycaemia, NICU admission, pre-eclampsia, shoulder dystocia |
| Silva[77] | 2012 | Brazil | 200 | 75 g OGTT fasting >5.3 or 1 hour >10.0 or 2 hours >8.0 | No screening | Apgar 5 min, BW, C-section, GA at birth, LGA, macrosomia, hypoglycaemia, NICU admission |

BW, birth weight; C-section, caesarean section; GA, gestational age; GDM, gestational diabetes mellitus; LGA, large for gestational age; NICU, neonatal intensive care unit.

adverse outcomes and for women diagnosed using more recently recommended criteria. Hence, we feel that it is important to place a moratorium on further small RCTs in this area and that funders should consider commissioning a multicentre large-scale RCT with adequate power to determine the effect and cost-effectiveness of different packages of care on adverse outcomes in women with GDM.

The evidence to support metformin use, although encouraging, has certain weaknesses. First, although there is a general 'trend' in favour of metformin use over insulin and glibenclamide (glyburide), CIs are wide, in both the direct and network meta-analysis comparing each two-way treatment effect. Second, the reporting of trial methods was generally poor with 'unclear or high risk of bias', and many trials included relatively few women and reported few outcomes. Third, in most trials directly comparing metformin with insulin, women receiving metformin were also given supplemental insulin 'if required'; in one of the largest trials, this equated to 46% of the metformin group.[59] Therefore, our results more appropriately relate to metformin's greater effectiveness as a first-line treatment for GDM rather than a standalone treatment compared with insulin.

In addition to being an effective first-line pharmacological treatment for GDM, metformin may also be preferred by women as it is administered orally and can be stored at room temperature, compared with insulin that requires subcutaneous injection and refrigerated storage. Metformin is sometimes associated with gastrointestinal upset, which may affect compliance and quality of life.

Few trials have reported side effects or measures of participant satisfaction or well-being, all important outcomes that have the potential to impact health and therefore should be evaluated. Recent guidance[16 44] recommends lower glucose thresholds compared with those previously recommended to diagnose GDM[39 40] (and used in the included trials). Therefore, it is possible that a greater proportion of women diagnosed with GDM will require only diet modification or less 'intensive' management compared with those previously diagnosed with GDM because their hyperglycaemia is less severe. There is a continuum of increasing risk of adverse outcomes across the spectrum of glucose however[80 81]; therefore, interventions to reduce hyperglycaemia even at lower glucose levels are likely to improve outcomes, but this needs confirming by large well-designed RCTs.

### Strengths and limitations
This systematic review and meta-analysis includes a large number of trials with varied populations and examines the effectiveness of treatment packages and diets as well as individual pharmacological treatments for reducing the risk of adverse perinatal outcomes.

For some comparisons, trials and numbers of women were few, as were outcomes reported. Trial quality was generally poor with subsequent high or unclear risk of bias. GDM diagnostic criteria varied across trials, and recently

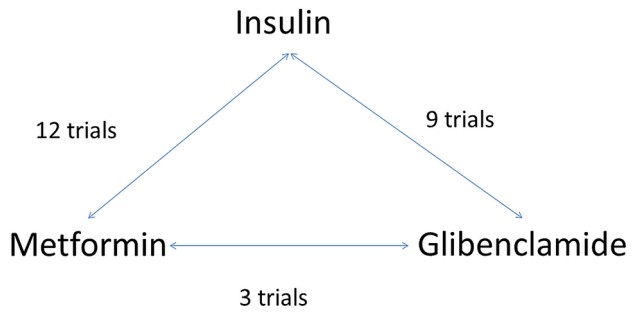

**Figure 3** Network meta-analysis, relationship of treatment comparisons.

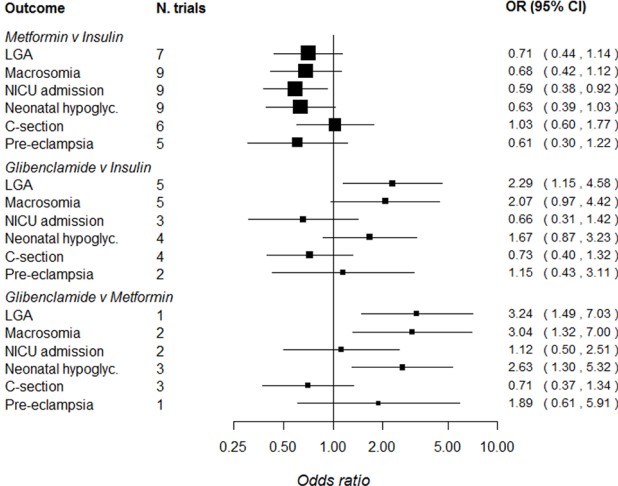

**Figure 4** Network meta-analysis comparing metformin, glibenclamide and insulin. First better, treatment listed first in the outcome column is superior; second better, treatment listed second in the outcome column is superior. C-section, caesarean section; LGA, large for gestational age; NICU, neonatal intensive care unit.

recommended thresholds are lower now compared with when most included trials were conducted.

Lower glucose threshold criteria recommended by the International Association of Diabetes and Pregnancy Study Groups[44] and subsequently endorsed by the WHO[82] aim to identify offspring at risk of obesity through its association with LGA (birth weight >90th percentile), cord C-peptide >90th percentile and percentage body fat >90th percentile. However, there are no trials that have used these criteria, and the classification of less severe hyperglycaemia when lower glucose thresholds are used to diagnose GDM may reduce the magnitude of the effect of interventions, compared with those reported by earlier trials using higher glucose thresholds. There has also been no longer term follow-up conducted to evaluate the treatment of GDM and the effects on risk of offspring outcomes. Importantly, few of the trials that we reviewed had reported side effects or measures of participant satisfaction or well-being.

### Implications for practice

This review provides reassurance that a package of care where a 'step up' approach of first providing dietary and lifestyle advice, then adding supplementary metformin or insulin if glucose levels are not adequately controlled, is a reasonable and effective approach compared with providing just routine antenatal care, particularly with

regard to reducing the risk of LGA. However, it has also highlighted the general poor quality of recent small RCTs that do not improve the evidence base but subject women with GDM to unnecessary 'experimentation' and are a cost to society.

Metformin seems to be an effective alternative to insulin, if diet modification inadequately controls hyperglycaemia; however, supplemental insulin may be required in up to 50% of women.[59] There is a need to cease further small RCTs in this area and conduct large well-designed RCTs that clarify the most effective treatment across a range of outcomes, including those that are likely to be important to women such as quality of life measurements and those identified by the Cochrane Pregnancy and Childbirth Group as being essential for trials and reviews of diabetes in pregnancy. These should be incorporated into current diagnostic criteria and ideally look at longer term outcomes in mothers and offspring.

**Correction notice** This paper has been amended since it was published Online First. Owing to a scripting error, some of the publisher names in the references were replaced with 'BMJ Publishing Group'. This only affected the full text version, not the PDF. We have since corrected these errors and the correct publishers have been inserted into the references.

**Acknowledgements** The authors thank Julie Glanville and Mick Arber of the York Health Economics Consortium, University of York, UK, for carrying out the searches.

**Contributors** DF, MS, DAL and TAS designed the study. MS wrote the statistical analysis plan. DF monitored the review process. DF, MS, MB, DAL, TAS and DT interpreted the data, DF, MS, MB and SG assessed studies for inclusion. MS cleaned and analysed the data. DF wrote the draft paper. All authors have approved the final version. DF is the guarantor and takes responsibility for the content of this article.

**Funding** This work was supported by the National Institute for Health Research (NIHR), Health Technology Assessment (HTA) programme, project number 11/99/02. DF holds a NIHR Post-doctoral Research Fellowship award (PD-2014-07-019). DAL works in a Unit that is supported by the University of Bristol and UK Medical

**Table 5** Estimated probability (%) of a treatment being the most effective in reducing the risk of a dichotomous outcome

| Outcome | Treatment | | |
| --- | --- | --- | --- |
| | Insulin | Metformin | Glibenclamide (glyburide) |
| Large for gestational age | 7.1 | 92.8 | 0.1 |
| Macrosomia | 5.6 | 94.0 | 0.3 |
| Neonatal intensive care admission | 0.5 | 61.2 | 38.3 |
| Neonatal hypoglycaemia | 3.3 | 96.3 | 0.4 |
| Caesarean section | 10.4 | 9.7 | 79.9 |
| Pre-eclampsia | 4.8 | 84.0 | 11.2 |

Research Council ((MC_UU_12013/5) and she holds a NIHR Senior Investigator award (NF-SI-0611-10196).The views and opinions expressed therein are those of the authors and do not necessarily reflect those of the HTA, NIHR, MRC, UK National Health Service (NHS) or the Department of Health.

**Competing interests** All authors have completed the ICMJE uniform disclosure form at www.icmje.org/coi_disclosure.pdf and declare: no support from any organisation for the submitted work; no financial relationships with any organisations that might have an interest in the submitted work in the previous three years; no other relationships or activities that could appear to have influenced the submitted work.

**Patient consent** Consent is not required when conducting a systematic review.

**Ethics approval** This study did not require ethical approval as the data used have been published previously, and hence are already in the public domain.

**Provenance and peer review** Not commissioned; externally peer reviewed.

**Data sharing statement** Extracted data are available upon request to the corresponding author.

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
