## [Reviewer comments · BMJ Open]

ARTICLE DETAILS

TITLE (PROVISIONAL)	Treatments for gestational diabetes: A systematic review and meta-analysis
AUTHORS	Farrar, Diane; Simmonds, Mark; Bryant, Maria; Sheldon, Trevor; tuffnell, derek; Golder, Su; Lawlor, Debbie

VERSION 1 - REVIEW

REVIEWER	Laura Gray University of Leicester, UK
REVIEW RETURNED	20-Dec-2016

GENERAL COMMENTS	Review of Farrar Treatments for gestational diabetes: A systematic review This paper summarises the results of a systematic review of treatments (both pharmacological and non-pharmacological) for gestational diabetes. Where appropriate pair-wise meta analyses are conducted and a network approach is used to compare pharmacological approaches. Although previous reviews have been completed in this area the authors clearly state how this review differs and build on the existing literature in this area. • Supplementary tables/figures were not included in the submission• I would like to see more detail regarding the NMA methodology – Variance structure, prior distribution used and assessment of model fit• Did you assess consistency in the NMA?• Why were OR pooled for the NMA but RR for the pairwise analysis?• Please include a network diagram for the NMA• For the continuous outcomes were data extracted from ITT analyses for all studies?• Make sure all acronyms are defined in the text (OGCT)
---

REVIEWER	Kerstin Berntorp Department of Endocrinology, Skåne University Hospital, Sweden
REVIEW RETURNED	03-Feb-2017

GENERAL COMMENTS	The aim of this systematic review was to investigate the effectiveness of different treatments for GDM and to determine which treatment was most effective. Only randomized controlled trials were included and results were pooled where appropriate. A network-analysis comparing all pharmacological treatments for GDM was done. It is concluded that packages of care are effective in reducing most adverse perinatal outcomes. However, it is underlined that trials are often small and poorly reported with unclear bias, and that large well-designed trials are urgently needed.
--

	This is a generally well written paper. The aim is relevant. The method is well described and appropriate. The results are presented in a relevant way. Discussion The results are discussed properly but the contribution of this study to new knowledge should be more emphasized. During the last couple of years there has been some additional reviews dealing with the topic that could be referred to, see below. What does this study add compared with these studies?  • Jiang YF et al. Comparative efficacy and safety of OADs in management of GDM: network meta-analysis of randomized controlled trials. J Clin Endocrinol Metab. 2015;100:2071-80. • Su DF, Wang XY. Metformin vs insulin in the management of gestational diabetes: a systematic review and meta-analysis. Diabetes Res Clin Pract. 2014;104:353-7. 19 • Butalia S et al. Short- and long-term outcomes of metformin compared with insulin alone in pregnancy: a systematic review and meta-analysis. Diabet Med. 2016 May 6. • Zhu B et al. Metformin versus insulin in gestational diabetes mellitus: a meta-analysis of randomized clinical trials. Ir J Med Sci. 2016;185:371-81. • Feng Y, Yang H. Metformin - a potentially effective drug for gestational diabetes mellitus: a systematic review and meta-analysis. J Matern Fetal Neonatal Med. 2016 Sep 9:1-8. The answer to the aims/study questions should be clearly stated in the abstract under conclusions. That treatment of GDM reduces adverse outcomes is generally accepted but which treatment is the most effective is still under debate, which I find the most important question to be answered. This should also be further stressed in the main text. Previous reviews have consistently shown that metformin is superior to glibenclamide (glyburide) and that metformin (with addition of insulin if needed) seems to be more effective than insulin alone. In spite of this, metformin is still not used in many countries. Lowering the diagnostic glucose thresholds according to the IADPSG recommendation will substantially increase the number of women diagnosed with GDM all over the world. Therefore, convenient cost-effective treatment options are desirable, such as the use of metformin as a first-line treatment with addition of insulin if required.
--	--

REVIEWER	Shakila Thangaratinam QMUL, London, UK
REVIEW RETURNED	13-Feb-2017

GENERAL COMMENTS	Originality: As the authors have mentioned, this question posed by the review has previously been addressed by other reviews. The current work is an update of evidence with increase in number of participants. Methods: The methodology is appropriate. The systematic review is well described. However, details are missing in the methodology and reporting of findings of network meta-analysis. For e.g..
--

	1. Why do the authors consider that they can make an indirect comparison where no head to head comparison exists. 2. The role of varied severity of GDM, on this assumption needs to be explained. While Women with mild GDM could be enrolled in an RCT of diet vs usual care, this might not be the case for metformin vs. insulin where women with severe GDM are recruited. 3. Need details on how they assessed the inconsistency or incoherence in the model. 4. The findings should provide details, a as Fig, on the network of studies. 5. No details are provided on method used for ranking of interventions. The main issue is the heterogeneity of the population, and intervention and these needs to be assessed through either subgroup or sensitivity analysis. Implications for clinical practice: Given the limitations in the findings, it is less likely to influence clinical practice.
--	--

VERSION 1 – AUTHOR RESPONSE

Reviewer: 1

This paper summarises the results of a systematic review of treatments (both pharmacological and non-pharmacological) for gestational diabetes. Where appropriate pair-wise meta analyses are conducted and a network approach is used to compare pharmacological approaches. Although previous reviews have been completed in this area the authors clearly state how this review differs and build on the existing literature in this area.

Supplementary tables/figures were not included in the submission

We have now included the risk of bias table (supplementary Table 1) which was omitted from our previous submission. All remaining Tables referred to in the manuscript are presented after the references at the end of the manuscript and Figures are presented as separate tif files (Figures 2a to 2h). For brevity we have included only forest plots related to combined trials and their effect estimates (with the exception of Supplementary figures 1 and 2 which show diet modification trial outcomes for each trial as they were too dissimilar to combine in a meta-analysis). However if the editors feel figures are required for each outcome and comparison we are happy to supply these. There are in excess of 50 figures.

- I would like to see more detail regarding the NMA methodology – Variance structure, prior distribution used and assessment of model fit

The NMA model was as originally set out by Lu and Ades.¹ It used a “Binominal-normal” structure, i.e. events were assumed to follow a binomial distribution, with log odds and random effects being normally distributed. Vague normal priors (mean 0, variance 10000) were used except for heterogeneity, where an inverse-gamma (0.1, 0.1) distribution was used. We have now included this information in the text.

- Did you assess consistency in the NMA?

Model fit and consistency were assessed by comparing the NMA results to the direct pairwise comparisons (in figures 2a to 2h). There is no evidence of inconsistency. We have now included this information in the text.

- Why were OR pooled for the NMA but RR for the pairwise analysis?

RRs are more usually used for direct treatment comparisons because of their more intuitive interpretation; however ORs were used for the NMA to ensure model stability, since log odds ratios would be expected to more closely follow a normal distribution. We have amended the manuscript to make this clear

- Please include a network diagram for the NMA

We have now included a diagram of the relationship of treatment comparisons for the NMA (Figure 3) and a forest plot of the network comparisons (figure 4)

- For the continuous outcomes were data extracted from ITT analyses for all studies?

All data was extracted according to an intention to treat analysis.

- Make sure all acronyms are defined in the text (OGCT)

We have amended our text to spell out oral glucose tolerance test (OGCT) in the first instance and proof read to find any similar errors

Reviewer: 2

The aim of this systematic review was to investigate the effectiveness of different treatments for GDM and to determine which treatment was most effective. Only randomized controlled trials were included and results were pooled where appropriate. A network-analysis comparing all pharmacological treatments for GDM was done. It is concluded that packages of care are effective in reducing most adverse perinatal outcomes. However, it is underlined that trials are often small and poorly reported with unclear bias, and that large well-designed trials are urgently needed.

This is a generally well written paper.

The aim is relevant.

The method is well described and appropriate.

The results are presented in a relevant way.

Discussion

The results are discussed properly but the contribution of this study to new knowledge should be more emphasized. During the last couple of years there has been some additional reviews dealing with the topic that could be referred to, see below. What does this study add compared with these studies?

- Jiang YF et al. Comparative efficacy and safety of OADs in management of GDM: network meta-analysis of randomized controlled trials. *J Clin Endocrinol Metab.* 2015;100:2071-80.
- Su DF, Wang XY. Metformin vs insulin in the management of gestational diabetes: a systematic review and meta-analysis. *Diabetes Res Clin Pract.* 2014;104:353-7. 19
- Butalia S et al. Short- and long-term outcomes of metformin compared with insulin alone in pregnancy: a systematic review and meta-analysis. *Diabet Med.* 2016 May 6.
- Zhu B et al. Metformin versus insulin in gestational diabetes mellitus: a meta-analysis of randomized clinical trials. *Ir J Med Sci.* 2016;185:371-81.
- Feng Y, Yang H. Metformin - a potentially effective drug for gestational diabetes mellitus: a systematic review and meta-analysis. *J Matern Fetal Neonatal Med.* 2016 Sep 9:1-8.

We had cited the first review (Jiang (2015)) in the discussion and have now included the remaining

citations (Butalia (2017), Feng (2016), Su (2014) and Zhu (2016)), all studies included in these reviews that met our inclusion criteria are in our review. We have made clearer in the text of the discussion that trials and reviews continue to be conducted that pay little attention to the non-pharmacological treatments for GDM. Given the changing characteristics of the population and the lower fasting diagnostic thresholds, it is important to understand how all treatments of GDM affect outcomes for these women, many of whom will not require pharmacological treatments. In the discussion we say that our pragmatic approach to evaluating the many trials examining treatment packages of care for women diagnosed with hyperglycaemia/GDM (which do acknowledge the influence of non-pharmacological treatments) makes our findings generalisable to most clinical situations. Our inclusion of 'all' treatments for GDM" along with the addition of newly conducted trials, a network meta-analysis and our interpretation of our findings is where our review differs from others.

The answer to the aims/study questions should be clearly stated in in the abstract under conclusions. That treatment of GDM reduces adverse outcomes is generally accepted but which treatment is the most effective is still under debate, which I find the most important question to be answered. This should also be further stressed in the main text. Previous reviews have consistently shown that metformin is superior to glibenclamide (glyburide) and that metformin (with addition of insulin if needed) seems to be more effective than insulin alone. In spite of this, metformin is still not used in many countries. Lowering the diagnostic glucose thresholds according to the IADPSG recommendation will substantially increase the number of women diagnosed with GDM all over the world. Therefore, convenient cost-effective treatment options are desirable, such as the use of metformin as a first-line treatment with addition of insulin if required.

We agree that treatment reduces the risk of several adverse outcome s, we have shown this in our analysis of treatment packages and we agree that even with the many trials included in our review it remains unclear which treatment is best, which is in part due to the methodological weaknesses of many of the trials. We have made this clearer in the abstract and the discussion. In the discussion we also suggest that the new lower thresholds recommended by the IADPSG may influence treatment effects because more women with less severe hyperglycaemia will be identified and that large well-designed and conducted trials are urgently needed (that use these lower thresholds).

Reviewer: 3

Originality: As the authors have mentioned, this question posed by the review has previously been addressed by other reviews. The current work is an update of evidence with increase in number of participants.

Methods: The methodology is appropriate.

The systematic review is well described. However, details are missing in the methodology and reporting of findings of network meta-analysis. For e.g..

1. Why do the authors consider that they can make an indirect comparison where no head to head comparison exists.

We note that direct head-to-head evidence exists for all three treatment comparisons (see Figure 3 for treatment comparisons). The direct evidence has been analysed using standard pairwise meta-analysis (see figures 2a to 2h) and these findings are consistent with the NMA findings.

As we described in the methods section, the purpose of a network meta-analysis is to allow synthesis by combining direct evidence from comparisons of treatments within trials and indirect evidence across trials on the basis of a common comparator, even when no direct comparison exists. We have

not included a resume of the strengths and weaknesses of network analysis in the discussion, however if the editors feel this would be beneficial we can provide this. The attributes of network analysis is widely discussed in the literature, and so we believe this is unnecessary and would detract from the aim of this paper. We have now included additional references in the methods section to direct the reader to further publications focusing on network meta-analysis (Higgins JPT, Thompson SG. Quantifying heterogeneity in a meta-analysis. *Stats Med.* 2002; 21(11): 1539-58. Bafeta A, Trinquart L, Seror R, Ravaud P. Reporting of results from network meta-analyses: methodological systematic review. *BMJ.* 2014; 348. Lumley T. Network meta-analysis for indirect treatment comparisons. *Stat Med.* 2002; 21(16): 2313-24. Song F, Altman DG, Glenny A-M, Deeks JJ. Validity of indirect comparison for estimating efficacy of competing interventions: empirical evidence from published meta-analyses. *BMJ.* 2003; 326(7387): 472. Mills EJ, Thorlund K, Ioannidis JP. Demystifying trial networks and network meta-analysis. *BMJ.* 2013; 346(f2914).)

2. The role of varied severity of GDM, on this assumption needs to be explained. While Women with mild GDM could be enrolled in an RCT of diet vs usual care, this might not be the case for metformin vs. insulin where women with severe GDM are recruited.

The severity of hyperglycaemia may influence the effectiveness of a treatment and we have added text to our discussion to suggest this, however although some trials do report subgroup (for example metformin/metformin with sup insulin) glycaemia at baseline, there is inconsistency, with some reporting a significant difference and others reporting similar median fasting and post-load glucose and others do not report differences by treatment subgroups.

Trials of treatment packages tend to recruit at GDM diagnosis, whereas pharmacological trials tend to recruit when diet and exercise have been ineffective at reducing hyperglycaemia. Unfortunately treatment package trials, including the most well conducted and largest trials^{2,3} do not report glycaemic levels at trial entry by treatment subgroup, so it is difficult to draw firm conclusions about whether women with the most severe hyperglycaemia generally require metformin or insulin and the extent to which this effects outcomes, because for the treatment subgroups (diet only, diet and insulin) there is a substantial risk of surveillance or detection bias, those receiving insulin being treated differently to those not (potentially monitored more closely, induced earlier etc) when there may be no clinical indication to do so. We have now added to the discussion regarding this issue.

3. Need details on how they assessed the inconsistency or incoherence in the model.

We believe the reviewer is referring to network inconsistency here. Inconsistency was assessed by comparison with the direct pairwise meta-analyses (Figures 2a to 2h compared with Figure 4). No evidence of inconsistency was identified. We have now made this clear in the manuscript

4. The findings should provide details, as Fig, on the network of studies.

We have now added a diagram of the relationship of treatment comparisons (Figure 3) and a forest plot showing the network comparisons (figure 4)

5. No details are provided on method used for ranking of interventions.

The main issue is the heterogeneity of the population, and intervention and these needs to be assessed through either subgroup or sensitivity analysis.

We believe the reviewer is referring to Table 5 with regard to ranking. This is a standard method, where the posterior probabilities of being most effective are calculated from the posterior odds as part of the Bayesian model as developed by Lu and Ades, we have added a sentence to the methods to make this clear. By definition women with GDM are a heterogeneous group (likely different causes for the hyperglycaemia and different effects from treatments). While subgroup or sensitivity analysis would be desirable, heterogeneity was found to be low or absent in most pairwise meta-analyses (see Figures 2a to 2h), and the number of trials are too few for subgroup analysis results to be reliable.

Implications for clinical practice: Given the limitations in the findings, it is less likely to influence clinical practice.

It is important that clinicians and researchers are fully aware of the evidence surrounding a treatment, including the strengths and limitations, so that informed decisions can be made. We have provided a comprehensive review of all treatments for GDM and assessed the attributes of the included trials and performed direct and indirect comparisons. Additionally we have made research recommendations and how the currently available evidence could influence clinical practise. Although there are limitations to this review, the consistency of our direct and indirect comparison results suggests our findings are valid, which is that our findings generally support the use of a 'step up' approach of firstly providing dietary and lifestyle advice, then adding supplementary metformin or insulin if glucose levels are not adequately controlled and that metformin seems to be an effective alternative to insulin.

References

1. Lu G, Ades AE. Combination of direct and indirect evidence in mixed treatment comparisons. *Stat Med.* 2004; 23(20): 3105-24.
2. Landon MB, Spong CY, Thom E, Carpenter MW, Ramin SM, Casey B, et al. A multicenter, randomized trial of treatment for mild gestational diabetes. *N Engl J Med.* 2009; 361: 1339-48.
3. Crowther CA, Hiller JE, Moss JR, McPhee AJ, Jeffries WS, Robinson JS. Effect of treatment of gestational diabetes mellitus on pregnancy outcomes. *N Engl J Med.* 2005; 352(24): 2477-86.

VERSION 2 – REVIEW

REVIEWER	Laura Gray] University of Leicester, UK
REVIEW RETURNED	09-Mar-2017

GENERAL COMMENTS	I am happy that my comments have been addressed,
--

REVIEWER	Kerstin Berntorp Department of Endocrinology Skåne University Hospital, Malmö, Sweden Lund University Sweden
REVIEW RETURNED	10-Mar-2017

GENERAL COMMENTS	The paper has improved after revision and the questions raised by the reviewers have been satisfactory responded to.
--